# Advances in Metabolic Engineering of *Pichia pastoris* Strains as Powerful Cell Factories

**DOI:** 10.3390/jof9101027

**Published:** 2023-10-19

**Authors:** Jian Zha, Dan Liu, Juan Ren, Zhijun Liu, Xia Wu

**Affiliations:** School of Food and Biological Engineering, Shaanxi University of Science and Technology, Xi’an 710021, China; ld18840338212@163.com (D.L.); 18791508437@163.com (J.R.); 18242054439@163.com (Z.L.)

**Keywords:** *Pichia pastoris*, CRISPR/Cas9, recombinant protein, natural product, cell factory

## Abstract

*Pichia pastoris* is the most widely used microorganism for the production of secreted industrial proteins and therapeutic proteins. Recently, this yeast has been repurposed as a cell factory for the production of chemicals and natural products. In this review, the general physiological properties of *P. pastoris* are summarized and the readily available genetic tools and elements are described, including strains, expression vectors, promoters, gene editing technology mediated by clustered regularly interspaced short palindromic repeats (CRISPR)/Cas9, and adaptive laboratory evolution. Moreover, the recent achievements in *P. pastoris*-based biosynthesis of proteins, natural products, and other compounds are highlighted. The existing issues and possible solutions are also discussed for the construction of efficient *P. pastoris* cell factories.

## 1. Introduction

Methylotrophic yeasts are microorganisms capable of utilizing methanol as the sole source of carbon and energy and are hence considered as potential hosts for green bio-manufacturing. Meanwhile, these microbes, when used as cell factories, have some other advantages such as high fermentation density, low accumulation of toxic metabolites, ability to grow in cheap basal salt media, and capability of complete protein modifications and processing.

*Pichia pastoris*, which has been renamed as *Komagataella phaffii* [1], is one of the typical methylotrophic yeasts and plays an important role in the production of recombinant proteins. Using the *P. pastoris* expression system, thousands of recombinant proteins have been synthesized so far, some of which have successfully entered the market, such as human insulin and human interferon-α [2].

*P. pastoris* is a unicellular microorganism with the advantages of high expression levels of recombinant proteins, ease of large-scale cultivation, and low cultivation costs [3]. Its cell density can reach 130 g/L during industrial production [4], and the yield of recombinant proteins can reach tens of grams per liter [5]. Recombinant proteins can be expressed both intracellularly and in a secreted form in *P. pastoris*. The efficiency of secretory expression in *P. pastoris* is generally higher than that in *Saccharomyces cerevisiae*, which is another commonly adopted workhorse for microbial bio-production [6]. *P. pastoris* has a clear genetic background and is excellent in protein folding and post-translational modifications, especially glycosylation [7,8]. In *P. pastoris*, oligosaccharide chains added to proteins after translation (8–14 mannose residues per side chain) are shorter than in *S. cerevisiae* (50–150 mannose residues), and *O*-linked glycosylation is very minimal, potentially avoiding the risk of excessive glycosylation (Figure 1) [9]. Moreover, *P. pastoris* contains a strong methanol utilization pathway and is able to provide some key cofactors [10]. Another major difference between these two yeasts is that the homologous recombination (HR) pathway in *P. pastoris* is significantly weaker than the non-homologous end joining (NHEJ) pathway [11]. In recent years, genetic and metabolic modifications of *P. pastoris* have been expanded and deepened attributed to gene editing tools, promoter engineering approaches such as promoter mutation libraries and synthetic core promoters, novel metabolic engineering strategies such as metabolic compartmentalization and cofactor engineering, and other techniques. Additionally, information on metabolic processes and strain physiology has gradually been revealed via systems biology and omics studies [12,13], which greatly benefits the biosynthesis of natural products in this organism. In this review, we introduce the physiological characteristics of *P. pastoris*, summarize the latest advances in this expression system, and list its practical applications in the production of recombinant proteins and value-added compounds.

## 2. Physiological Characteristics of *Pichia pastoris*

*P. pastoris* belongs to the Ascomycetes class [14]. Its colony is generally milk-white with a smooth surface, showing a bulge (Figure 2). This yeast mainly exists in the haploid form in the asexual growth phase. Under nutrient limitation, two haploid cells can be induced with different physiological types to mate and fuse into diploids [15].

*P. pastoris* grows optimally at 28–30 °C [16] with a tolerance of pH ranging from 3 to 7 [17]. The carbon source includes a variety of compounds such as glucose [18], glycerol [19], L-rhamnose [20], formate [21], and so on. This microbe can also use methanol as its sole source of carbon and energy owing to the presence of enzymes in the peroxisomes essential to methanol metabolism, such as alcohol oxidase, dihydroxyacetone synthase, and peroxidase [22]. Among these, the expression of alcohol oxidase is strictly induced and controlled by methanol, so the enzyme activity can only be detected in the presence of methanol [23]. *P. pastoris* can use ammonium sulfate, proline, and peptone as the nitrogen source. When ammonium sulfate is adopted, the expression levels of major genes related to methanol utilization (gene *MUT*) and peroxisome biogenesis/degradation (gene *PEX*) are the highest [24]. Meanwhile, an appropriate NH_4_^+^ concentration is beneficial for cell growth and heterologous protein expression [24,25]. In some cases, the use of complex nitrogen sources such as casamino acids and peptone can effectively alleviate proteolytic degradation of heterologous proteins [26]. 

## 3. The *Pichia pastoris* Expression System

### 3.1. Strains

At present, the commonly used *P. pastoris* strains for heterologous protein expression include X-33, GS115, KM71, SMD1163, and MC100-3. The characteristics of these strains have been described in detail in other reports [27,28]. Based on their ability to utilize methanol, *P. pastoris* strains are mainly divided into three phenotypes, i.e., Mut^+^, Mut^S^, and Mut^−^. Strain X-33 is a wild-type strain carrying genes *AOX1* and *AOX2* (both encoding alcohol oxidase) and the phenotype is Mut^+^ (methanol utilization plus), which is often used to express recombinant plasmids containing zeocin resistance. Both GS115 and KM71 strains are histidine deficient. The GS115 strain contains *AOX1* and *AOX2*, and its own phenotype is Mut^+^. In comparison, the KM71 strain is Mut^S^ (methanol utilization slow) because the gene *AOX1* is replaced by the *S. cerevisiae Arg4*, and the strain can only utilize methanol at a slower rate dependent on the weakly controlled gene *AOX2*. Strain MC100-3 is deficient in both *AOX* genes and hence cannot grow on methanol, thus performing the Mut^−^ (methanol utilization minus) phenotype. SMD1163 is a protease deficient strain in which the *pep4* gene encoding protease A and the *prb1* gene encoding a subtilisin-like protease are knocked out. This strain is favorable for the expression of heterologous proteins sensitive to proteases [29]. 

Dozens of *P. pastoris* strains, including X-33 and GS115, have been compared genomically and transcriptomically to analyze why recombinant protein expression in these strains is different [30]. Genomic analysis shows that there are functional and non-functional single-nucleotide polymorphisms in the genome sequences of different strains, which affect DNA repair, cell cycle, cell wall structure and so on. Transcriptomic analysis shows that there are variations in the expression levels of genes involved in the pentose phosphate pathway, methanol utilization pathway, and other pathways in different strains under different growth conditions. Compared with other strains, the higher transformation efficiency and better protein secretion for strains X-33 and GS115 may be due to the higher permeability of their cell wall, which has been a focus of current studies [30].

### 3.2. Expression Vectors

According to the location of the expressed proteins, *P. pastoris* vectors can be divided into intracellular expression vectors (such as pPIC3, pPICZ, pPHIL-D2, etc.) and secretory expression vectors (such as pPIC9, pPIC9K, pPICZα, etc.), in which the secretory expression vector usually contains a signal peptide sequence inserted behind the promoter. The characteristics of the expression vectors commonly used at present are summarized in detail in a recent report [31].

Integrative plasmids are frequently used for exogenous gene expression in *P. pastoris*. Such vectors are typically constructed as *E. coli*/*P. pastoris* shuttle vectors. On the one hand, these vectors contain elements for plasmid amplification in *E. coli*, including an origin of replication and a selection marker, which is usually antibiotic resistance. On the other hand, these vectors carry components required for heterologous gene expression in *P. pastoris*, including the promoter/terminator, the multiple cloning site, and a proper selection marker. The selection marker can be an auxotrophic marker (HIS4, ARG4, URA3, etc.) or an antibiotic resistance marker (zeocin, G418, etc.).

To introduce a foreign gene into *P. pastoris*, episomal vectors are generally accompanied with easier manipulation and higher efficiency compared with gene integration into the chromosome. However, natural autoreplicative vectors have limited applications in *P. pastoris* due to their instability and the uneven distribution among progeny cells during cell division. To solve these issues, the genome of *P. pastoris* has been analyzed and the chromosome-2 centromeric DNA sequence has been identified to facilitate stable autoreplication and accurate distribution of the plasmid [32]. The stability of the episomal vectors can also be improved by introducing the autonomously replicating sequences (ARS) from other organisms, such as the panARS derived from *Kluyveromyces lactis* [33].

### 3.3. Promoters

#### 3.3.1. *AOX1* Promoter

Upon entry into *P. pastoris*, methanol is decomposed into formaldehyde and hydrogen peroxide by alcohol oxidase encoded by *AOX1* and *AOX2*. However, the enzyme activity is mostly provided by *AOX1* due to the low expression level of *AOX2* [23]. The endogenous promoter controlling *AOX1* expression, i.e., P*_AOX1_* upstream of *AOX1*, can be efficiently induced by methanol at low concentrations, with the optimal concentration ranging from 0.5% to 2.0% [6].

In recent years, with the continuous mechanistic research on the transcriptional regulation of P*_AOX1_*, the cis-acting elements and transcription factors that affect the transcriptional activity of this promoter have been discovered. Inan et al. divided P*_AOX1_* into five segments (A~E), of which fragments B and E play a promotional role for *AOX1* expression (Figure 3A) [34]. In *P. pastoris*, the response of P*_AOX1_* to methanol is positively regulated by a cascade of transcription factors including Mit1, Mxr1, and Prm1, which bind to different sites of P*_AOX1_* [35]. Among them, Mxr1 is closely related to the carbon-source-induced repression of P*_AOX1_*. When methanol is used as the carbon source, Mxr1 is transferred from the cytoplasm to the nucleus, leading to the derepression of P*_AOX1_*. At this point, Prm1 activates its own expression and the expression of Mit1, thus inducing strong activation of P*_AOX1_*. The strength of P*_AOX1_* can be improved via Mit1 overexpression [36]; however, Mit1 exerts feedback inhibition of Prm1 expression (Figure 3B) [37]. In consequence, the expression level of Mit1 needs to be delicately tuned to achieve the optimal expression of genes controlled by P*_AOX1_*. 

#### 3.3.2. GAP Promoter

Inducible promoters generally have some intrinsic disadvantages, such as costs and toxicity associated with the inducers and concerns of leaky expression of the target genes. A preferred alternative is the constitutive promoter. Genes regulated by constitutive promoters can be normally expressed during cell growth without the involvement of inducers. In *P. pastoris*, the most commonly used constitutive promoter is the glyceraldehyde-3-phosphate dehydrogenase promoter (P*_GAP_*), which is considered as a standard promoter for methanol-free expression systems [38,39]. This promoter is regulated by the metabolism of carbon sources, and its transcription level is the highest in the presence of glucose and is the lowest when cells are fed with methanol [40]. In a study on the recombinant expression of renal peptide transporter rPEPT2 in *P. pastoris* using P*_GAP_*, the expression level of rPEPT2 was about 5-fold higher in glucose than in methanol [41].

P*_GAP_* is generally weaker than P*_AOX1_* and attempts have hence been made to improve the strength of P*_GAP_*. Ata et al. [42] analyzed the transcription factor binding sites of P*_GAP_*. Via targeted deletion or overexpression of these sites, a promoter library was constructed with different strength in initiating gene expression. Moreover, a GAL4-like transcription factor was found to be critical in regulating the strength of P*_GAP_* and its overexpression could significantly improve the production of heterologous proteins controlled by P*_GAP_* [42]. This transcription factor is homologous to the GAL4 transcription factor in *S. cerevisiae*, which is involved in the regulation of galactose utilization [43].

#### 3.3.3. Other Promoters

Although the use of methanol as the sole source of carbon and energy is a major advantage of the *P. pastoris* cell factories, the toxicity and safety issues of methanol impose many limitations on practical industrial applications [44]. To solve this problem, methanol-free induction systems have been developed based on P*_AOX1_*. Chang et al. synthesized a positive feedback circuit in which P*_AOX2_*-driven Mxr1 promotes the transcription of P*_AOX1_*, while P*_AOX1_* is induced under glycerol starvation or in the absence of carbon sources [45]. Kinases have been proposed as potential targets for regulating the repression of P*_AOX1_* in the presence of a common carbon source such as glycerol [46]. By targeting the genes encoding glycerol kinase (*gut1*) and dihydroxyacetone kinase (*dak*), non-methanol-inducible P*_AOX1_* expression systems were constructed using glycerol and dihydroxyacetone as carbon sources to induce P*_AOX1_* expression, respectively, although the induction was not as efficient as that induced by methanol [47].

In addition to the above modifications of targets or transcription factors involved in the regulation of *P_AOX1_*, a large number of novel promoters suitable for the *P. pastoris* expression system has also been explored [48,49,50], with the recently identified promoters listed in Table 1. Among these, the promoter P*_ADH2_* is naturally present in *P. pastoris* to regulate the expression of the alcohol dehydrogenase gene *ADH2*, which is involved in the conversion of ethanol to formaldehyde. This promoter can be activated via direct interaction with the transcription factor Mxr1 [51]. By replacing the repressor region of P*_ADH2_* with the activator region, the *SNT5* promoter is obtained, which is much stronger than P*_ADH2_* [51].

*P. pastoris* can use L-rhamnose as a sole carbon source, and its metabolism is heavily dependent on L-rhamnonate dehydratase and L-2-keto-3-deoxyrhamnonate aldolase encoded by *LRA3* and *LRA4*, respectively [20]. The relevant endogenous promoters of these genes, i.e., P*_LRA3_* and P*_LRA4_*, are tightly regulated by L-rhamnose, among which P*_LRA3_* has a priming strength equivalent to that of P*_GAP_* and can efficiently drive the production of foreign proteins [20].

#### 3.3.4. Synthetic Core Promoter Engineering

The core promoter region plays a pivotal role in the regulation of gene expression, and its genetic modification is an important content of promoter engineering. The core promoter is the region necessary for RNA polymerase to recognize and initiate transcription, and it consists of the RNA polymerase binding site, the TATA box, and the transcription start site. In *P. pastoris*, fully synthetic core promoters and the 5′-untranslated region have been designed and applied to P*_AOX1_*, resulting in a series of promoter libraries with different expression levels [59]. Portela et al. designed and synthesized 112 synthetic core promoter sequences according to the sequence and function relationship, nucleosome occupancy, and the existence of short motifs of the natural core promoter. The synthetic sequences were fused with the cis-regulation module upstream of *P. pastoris AOX1*, which significantly improved the activity of the promoter. In addition, these synthetic sequences could be used interchangeably for different core promoters without affecting the activity [60]. 

### 3.4. Signal Peptides

Secretory expression of proteins is generally dependent on cleavable signal peptides (usually 15–50 amino acids) that direct the transmembrane transfer of the newly synthesized peptides and proteins. These signal peptides, mostly located at the N-termini of the secreted proteins, usually contain three domains, i.e., the positively charged basic N-terminus (1–5 amino acids), the hydrophobic center that forms a helical structure (7–15 amino acids), and the highly polar C-terminus (3–7 amino acids) that serves as the cleavage site [61]. With a great impact on the extent of protein folding and the rate of protein secretion, signal peptides play a crucial role in high-level expression and secretion of functional proteins [62]. 

The signal peptides commonly used for protein secretion in *P. pastoris* include the signal sequence of α-factor and invertase-2 (SUC2) of *S. cerevisiae*, and *P. pastoris* acid phosphatase signal peptide (PHO1). Among these, α-factor is used most frequently and is mainly suitable for the secretory expression of peptides and small proteins. To further improve the efficiency of secretion, this signal peptide has been engineered through codon optimization, modification of the hydrophobic region, addition of spacer sequences, and site-directed mutagenesis [63,64,65]. Recently, four novel endogenous signal peptides have been discovered, including Dan 4, Gas 1, Msb 2, and Fre 2, according to the reported secretomes and genomes of *P. pastoris* [66]. This greatly expands the pool of signal peptides to be selected for suitable expression and secretion of recombinant proteins in *P. pastoris*. In addition, several signal peptides naturally present in other organisms can also be used for the expression of exogenous proteins in *P. pastoris*. Examples include the recombinant expression of human lysozyme in *P. pastoris* using the signal peptide from human serum albumin [67], and the secretory expression of *Candida antarctica* lipase B in *P. pastoris* guided by the *C. antarctica* lipase B signal peptide, with a higher secretion efficiency than that of α-factor [68]. However, given its importance, it is not easy to predict which signal peptide results in the best expression and secretion of the protein product, and different secretion efficiency can be achieved even in the same strain using the same signal peptide for distinct heterologous proteins.

### 3.5. CRISPR/Cas9 Genome Editing in Pichia pastoris

Since *P. pastoris* is an important workhorse for the synthesis of various bio-products, it is crucial to establish efficient and concise gene editing technologies for the genetic modifications of this microbe. Traditionally, gene insertion/deletion/replacement of *P. pastoris* relies on homologous recombination, which is inefficient with a low rate of success even when long homologous arms (sometimes more than 1 kb) are used. This is due to the domination of NHEJ over homologous recombination in this yeast [11]. Deletion of *KU70* impairs NHEJ and significantly facilitates homologous recombination at the expense of a lower transformation efficiency and slower cell growth.

CRISPR/Cas9 introduces breaks in DNA sequences complementary to the sgRNA, which are then repaired by host cells. Thereby, genetic modifications can be introduced programmably at desired locations by using sgRNA with particularly designed sequences. Compared with traditional homologous recombination-guided genomic modification, CRISPR/Cas9 is highly flexible in the sense that only sgRNA needs to be re-designed for each independent genomic modification process. As one of the most potent and convenient gene editing technologies, CRISPR/Cas9 has been explored extensively in the engineering of *P. pastoris*, as reviewed elsewhere [8,69,70].

CRISPR/Cas9-mediated genomic editing in *P. pastoris* relies on the correct expression of Cas9 and sgRNA in the nucleus, which can be affected by a series of factors. To improve the expression, researchers used RNA Pol II promoter for sgRNA expression, added ribozyme sequences both upstream and downstream of sgRNA, and optimized the codon of Cas9. After such optimization, near 100% efficiency could be achieved in *P. pastoris* for gene deletion, and multiplex gene deletion and targeted gene insertion was achieved efficiently with the aid of NHEJ [71]. This system was further introduced into a *KU70*-knockout strain, so that DNA breaks could be repaired through homologous recombination. Despite lower cell viability, near 100% efficiency of gene integration was achieved [72]. Recently, it has been reported that high-level expression of *RAD52* (radiation sensitive 52 that limits homologous recombination repair) helps to increase the single-gene editing efficiency to 90%; additionally, deletion of the *MPH1* gene (mutator phenotype encoding a member of the DEAH family of proteins) can improve the efficiency of multi-fragment recombination by 13.5-fold [11]. Meanwhile, based on this efficient genome editing platform, 46 neutral sites have been identified for heterologous gene integration under various growth conditions, where the heterologous gene could be placed without affecting the basic cell metabolism [11]. 

The biosynthetic pathways of value-added compounds (natural products and bulk chemicals) are often complex and involve multiple pathway genes. Therefore, a competent genetic tool that can manipulate multi-gene pathways has important implications for the application of *P. pastoris* as a cell factory. Based on the *KU70* knockout strain, a CRISPR/Cas9-mediated marker-less multi-site gene integration method has been developed, for which various sgRNA targets are designed within 100 bp upstream of the promoter or downstream of the terminator. Using this method, the integration efficiency of double-locus could reach 57.7%–70% [73]. The same method was used to establish a standardized CRISPR-based synthetic biology toolkit, in which the integration efficiency of double-locus could reach ~93% [74]. This toolkit allowed for one-step assembly of the biosynthetic pathways of 2,3-butanediol, *β*-carotene, zeaxanthin, and astaxanthin [74]. Meanwhile, efficient genome integration of heterologous genes has been investigated with short homology arms. The recombination mechanism of *S. cerevisiae* was introduced into *P. pastoris* by overexpressing genes related to the *S. cerevisiae* homologous recombination (*RAD52*, *RAD59*, *SAE2*, etc.), and a final 98% efficiency of double-locus integration was achieved using a homologous arm of ~40 bp [75]. 

Most of the existing forms of the CRISPR/Cas9 system in *P. pastoris* are plasmids. However, the stability and copy number of the plasmid have a certain impact on the efficiency of gene editing. Therefore, researchers have been attempting to identify replicons with high stability and appropriate copy numbers for improving the editing efficiency of the CRISPR/Cas9 system. For example, Gu et al. constructed a set of episomal plasmids containing autonomous replication sequences of different species and systematically compared their differences in transformation efficiency, copy number, and stability. The plasmid stability could be significantly enhanced when the replication origin of the plasmid, i.e., PARS1, was replaced with panARS from *Kluyveromyces lactis*, and the gene editing efficiency of this CRISPR/Cas9 system increased by up to 10-fold [33].

## 4. Adaptive Laboratory Evolution of *Pichia pastoris*

Adaptive laboratory evolution (ALE) is a method to artificially simulate the mutation and selection process in natural evolution under laboratory conditions such that the directed evolution of microorganisms can be achieved within a short period of time and mutated microbes with desired traits can be screened [76]. Compared with metabolic engineering, ALE only focuses on the generation of appropriate interference factors without detailed information on the intricate and intersecting metabolic networks, thus demonstrating broad applicability and strong practicability. ALE is one of the most effective methods of strain construction toward high-level synthesis of bio-products [77,78]. Although widely used in *S. cerevisiae* and *E. coli*, ALE only has limited applications in *P. pastoris*, and there is still a large space for development.

Efficient use of carbon sources and substrates is key to the high-level microbial production of bio-products, and ALE has been adopted in promoting the metabolic performances of *P. pastoris* on various nutrients or substrates. Moser et al. investigated the effect of growth media on cell growth and recombinant protein production in *P. pastoris* X-33 using methanol as a carbon source for continuous subculture in eutrophic medium YPM and low-nutrient medium BMM. After approximately 250 generations, evolved strains showed higher growth rates. Whole genome sequencing identified mutations in the *AOX1* gene involving the methanol binding region and its vicinity, leading to, surprisingly, a decline in AOX activity, possibly due to less intracellular accumulation of the toxic compound formaldehyde. Such methanol adaptation led to significantly higher titers of recombinant human serum albumin and fused lobes hexosaminidases [79]. Similarly, adaptation of *P. pastoris* GS115 toward xylose utilization in a sequential batch culture improved the consumption rate of xylose after 50 generations of evolution, and the evolved strain could consume 18 g/L of xylose in 72 h, which was 25% higher than the initial strain [80].

*P. pastoris* is naturally a heterotrophic microbe. Excitingly, scientists converted it into an autotrophic yeast using CO_2_ as the sole carbon source by replacing the peroxisomal methanol utilization pathway with the CO_2_ fixation pathway [81]. However, the engineered strain exhibited a specific growth rate of 0.008 h^−1^, which was far from actual requirements. To improve cell growth, ALE was conducted using serial batch cultivations in the presence of CO_2_ and methanol, and the evolved strain exhibited a specific growth rate of 0.018 h^−1^. Further analysis of the evolved strain identified that the mutation of *Nma1* (encoding nicotinic acid mononucleotide adenylyltransferase) and *PRK* (encoding phosphoribulokinase) genes reduced the activity of the relevant enzymes and promoted the intracellular ATP levels, thus leading to an increase in the growth rate [82].

## 5. Practical Applications of *Pichia pastoris* as a Cell Factory

Since Philips Petroleum Company released the *P. pastoris* expression system to academic research laboratories in 1993, the expression system has developed rapidly [83]. *P. pastoris* has gradually replaced *S. cerevisiae* as the eukaryotic expression system because it secretes very few endogenous proteins and has glycosylation similar to that in mammalian cells. *P. pastoris* has thus been gradually developed into a common host for the expression of medical and industrial enzymes, and thousands of recombinant proteins have been successfully produced [84]. In addition, natural products with diverse structures have also been synthesized in this host (Figure 4 and Table 2), which overcomes the disadvantages associated with chemical synthesis or extraction from plants that are traditionally used for their production. These accomplishments have promoted the engineering and development of *P. pastoris* as a potent and potential microbial platform [8].

### 5.1. Recombinant Proteins

#### 5.1.1. Nanobodies

Nanobodies, the natural antibodies first found in the serum of camels and sharks, are the smallest units known to bind antigens [100,101]. Compared with traditional antibodies, nanobodies have unique properties such as strong antigen binding, low immunogenicity, high solubility and stability, and low molecular weight, which offer potential advantages in disease diagnosis and treatment [102]. For example, nanobody neutralization therapy has been employed in the treatment of the coronavirus COVID-19 [103]. Nanobodies can be stably expressed in *P. pastoris* besides prokaryotic hosts [104]. For example, nanobodies against *Clostridium botulinum* neurotoxin were expressed in *P. pastoris*, and the yield and quality obtained were higher than those produced by bacteria [105]. An antitumor nanobody was functionally expressed in *P. pastoris*, and the yield and the half-life were improved via fusion with human serum protein [106]. 

To avoid some of the drawbacks of methanol-induced systems in *P. pastoris*, constitutive expression for nanobody production has been developed. Chen et al. constructed the constitutive anti-CEACAM5 nanobody expression system under the control of the *GAP* promoter and finally obtained 51.71 mg/L of the nanobody in a shake flask with process optimization [107]. This was the first report on the expression of nanobodies under a constitutive promoter in *P. pastoris*, which lays the foundation for the constitutive synthesis of other nanobodies. The *GAP* promoter was also adopted for the functional expression of other nanobodies in *P. pastoris* [108]. 

#### 5.1.2. Human Proteins

Proteins are essential components of human tissues and participate in a variety of essential physiological activities in the body such as maintenance of the normal metabolism and transport of various substances across membranes. Abnormal expression of proteins in the human body may cause various diseases. Therefore, these proteins become effective targets for disease treatment, and it is necessary to construct heterologous expression platforms to synthesize a large number of humanized recombinant proteins for drug screening. 

Human serum albumin (HSA) is the major protein in human plasma and is widely used for drug delivery. Its recombinant expression in *P. pastoris* GS115 has been reported, and a high yield of 8.86 g/L was obtained after process optimization [109]. Human coagulation factor XII plays an important role in thrombosis, and its abundant supply is necessary for inhibitor screening in the development of antithrombotic drugs. The recombinant serine protease domain of human coagulation factor XII has been expressed in *P. pastoris* X-33 with a yield of 20 mg/L and a clotting activity similar to that of its natural counterpart [110]. Compared with these cellular proteins which can be expressed easily in the soluble form, high-level microbial expression of human membrane proteins can be a real challenge. Nonetheless, a human multichannel membrane protein named sterol ∆8-∆7 isomerase has been successfully expressed in the form of a GFP fusion in *P. pastoris* as well as in *E. coli* and *S. cerevisiae*, with the best expression achieved in *P. pastoris* at 200 mg/L in shake flasks and 1000 mg/L in condensed culture [111]. Insulin is a typical drug for the treatment of diabetes. For its expression, the codon-optimized gene encoding insulin precursor (IP) fused to the α-factor signal peptide was integrated into the genome of *P. pastoris* X-33. IP was produced at 3 g/L in batch fermentation using a medium containing 2 g/L methanol at a low salt and high glycerol concentration. Recombinant human insulin with a purity of 99% was obtained after IP processing and transpeptidation [112]. Further optimization of the fermentation and purification processes greatly saved time and economic costs, facilitating fast recovery of recombinant human insulin [113]. In the case of the expression of recombinant human interferon α 2b (huIFNα2b) in *P. pastoris*, a production titer of 436 mg/L was achieved in a 1.7 L bioreactor with medium optimization guided by the design of experiments and artificial intelligence, and the key components of the optimal medium contained 46 g/L glycerol, 10 g/L ammonium sulfate, and 1.38% (*v*/*v*) methanol. This protein was expressed with human-type *N*-glycosylation and presented anti-proliferative activity on breast cancer cells [114].

### 5.2. Value-Added Compounds

#### 5.2.1. Terpenoids

Terpenoids are secondary metabolites with isoprene as the basic structural unit, and they mainly include monoterpenoids, sesquiterpenoids, diterpenoids, triterpenoids, and polyterpenoids [115]. Most of these compounds have anti-tumor, anti-inflammatory, and immunomodulatory effects, and are widely used in food processing and pharmaceutical manufacturing industries [116]. Yeast cells generally produce more precursors than bacteria for terpenoid biosynthesis, including DMAPP (dimethylallyl diphosphate) and IPP (isopentenyl diphosphate) [117]. Compared with other yeast chasses, *P. pastoris* can reach a higher fermentation density, produce fewer metabolic byproducts, and present strong tolerance to complex environments. These traits make *P. pastoris* suitable for terpenoid biosynthesis. 

Currently, a variety of common terpenoids such as astaxanthin and *β*-carotene has been produced in *P. pastoris*, with great breakthroughs achieved recently in the biosynthesis of some other terpenoid compounds such as *α*-santalene, catharanthine, lycopene, and *α*-farnesene. The biosynthesis of *α*-santalene in *P. pastoris* was achieved for the first time by integrating the codon-optimized santalene synthase gene *SAS* into the chromosome of *P. pastoris* using CRISPR/Cas9. The production was elevated via promoter optimization for *SAS* expression, overexpression of key genes (*tHMG1*, *IDI1*, and *ERG20*) in the mevalonate (MVA) pathway, and by increasing the copy number of *SAS*. Combined with process optimization, the production titer of *α*-santalene reached 21.5 g/L in fed-batch fermentation [85]. 

The monoterpene indole alkaloid catharanthine is a precursor of the potent anticancer drug vincristine. This plant natural product is expensive with limited supply due to its complex structure and low abundance. Its de novo biosynthesis from methanol and mannitol has been achieved in *P. pastoris* via division of the entire pathway into three modules, i.e., the nepetalactol module, the strictosidine module, and the catharanthine module. These modules were integrated into the genome at optimized sites and the pathway enzymes were screened for activity and selectivity. Committed steps were identified and the corresponding genes were overexpressed with higher copy numbers. Combined with process optimization, catharanthine was produced at 2.57 mg/L in fed-batch fermentation, representing the most complicated molecule heterologously synthesized in a non-model microbe [86].

Lycopene is a kind of carotenoid with high nutritional value and its biosynthesis in engineered *P. pastoris* has been reported with a relatively low yield [118]. To improve the production, a recent study integrated the lycopene biosynthetic gene from *Corynebacterium glutamicum* ATCC 13032 into the genome of *P. pastoris* GS115 via homologous recombination under the control of P*_AOX1_*. Precursor supply was enhanced via genomic integration of an extra copy of *HMGS* and *HMGR*, which are critical genes in the MVA pathway. Since the generation of GGPP (geranylgeranyl diphosphate) catalyzed by GGPPS (geranylgeranyl diphosphate synthase) is a rate-limiting step, the expression of the endogenous *GGPPS* gene was up-regulated by integrating two copies of this gene into the genome. These efforts led to a lycopene yield of 6.146 mg/g dry cell weight in shake flasks, which is the highest for lycopene production in *P. pastoris* [87]. 

*α*-Farnesene is a volatile sesquiterpene with important applications in the fuel, food, and pharmaceutical industries. Its biosynthetic pathway mainly includes three modules, i.e., the acetyl-CoA biosynthetic pathway, the MVA pathway, and the terminal biosynthetic pathway of *α*-farnesene. In a recent study, the production of this compound in *P. pastoris* was promoted after an array of engineering efforts, including (i) episomal overexpression of key genes involved in the MVA pathway and *α*-farnesene biosynthesis, i.e., *HMG1*, *IDI1*, and *ERG20*; (ii) introduction of ATP-dependent citrate lyase (ACL) from *Yarrowia lypolitica* and pyruvate dehydrogenase (cytoPDH) from *E. coli* MG1655 for a higher level of cytoplasmic acetyl-CoA; (iii) introduction of the isopentenol utilization pathway (IUP) into *P. pastoris* peroxisomes for better accumulation of IPP and DMAPP, and (iv) co-feeding of sorbitol and oleic acid as carbon sources. These attempts led to the synthesis of 2.56 g/L α-farnesene [119]. The synthesis of α-farnesene usually requires six molecules of NADPH and nine molecules of ATP. To further improve the production, the biosynthetic pathway of these two cofactors was reprogrammed via overexpression of key enzymes in the oxidative pentose phosphate pathway and introduction of the *S. cerevisiae*-derived NADH kinase POS5, which catalyzes the formation of NADPH from NADH. The supply of NADPH and ATP was optimized by increasing AMP accumulation required for ATP synthesis and decreasing NADH consumption. Such engineering resulted in a yield of 3.09 g/L [88].

#### 5.2.2. Polysaccharides

Polysaccharides widely exist in nature and participate in important cellular processes as an energy source and organizational structure. Polysaccharides can be used as efficient drug carriers due to their biodegradability, biocompatibility, and generally low costs [120]. Polysaccharide biosynthesis using engineered microorganisms has been attracting increasing attention, with some of the focus laid on *P. pastoris* due to its abundant supply of sugar precursors.

Hyaluronic acid (HA) is a polysaccharide with high clinical value, especially the high molecular weight HA. Microbial production of high molecular weight HA has been limited by insufficient precursor supply and cell growth inhibition [121]. To tackle this problem, the *Xenopus laevis xhasA2* and *xhasB* genes (encoding hyaluronan synthase 2 and UDP-glucose dehydrogenase, respectively) were expressed in *P. pastoris*, and 1.2 MDa HA was successfully synthesized upon overexpression of the endogenous genes encoding UDP-glucose pyrophosphorylase (*hasC*), UDP-*N*-acetylglucosamine pyrophosphorylase (*hasD*), and phosphoglucose isomerase (*hasE*) [89]. Further engineering work adopted the weak *AOX2* promoter in replacement of the strong *AOX1* promoter to reduce the expression of hyaluronan synthase. Such an attempt increased the molecular weight of HA to 2.5 MDa with a yield of 0.8–1.7 g/L at 26 °C to alleviate growth inhibition [89].

Chondroitin sulfate is one of the main drugs for the prevention and treatment of arthritis. Microbial synthesis represents a green technology compared with extraction from animal tissues. Recently, researchers constructed the biosynthetic pathway of chondroitin sulfate in *P. pastoris* via a series of engineering efforts [90]. First, *kfoC* and *kfoA* from *E. coli* K4 and *tuaD* from *Bacillus subtilis* were introduced into *P. pastoris* and codon-optimized, resulting in 189.9 mg/L of chondroitin. Then, expression of active chondroitin-4-*O*-sulfotransferase was achieved via optimization of the promoter and the Kozak sequence, which led to the production of 182.0 mg/L of chondroitin sulfate with a sulfation degree of 1.1%. Next, the sulfation degree was elevated to 2.8% by overexpressing adenosine-5′-triphosphate sulfurylase and adenosine 5′-phosphosulfate kinase to enhance the supply of 3′-phosphoadenosine-5′-phosphosulfate. These efforts led to the production of 2.1 g/L of chondroitin sulfate from methanol with a sulfation degree of 4.0% in fed-batch fermentation.

*P. pastoris* can synthesize short- and low-immunogenic humanized glycosyl chains for attachment to recombinant glycoproteins, and are thus considered suitable for the biosynthesis of oligosaccharides. 2′-Fucosyllactose (2′-FL) is a major oligosaccharide in breast milk, and its application in infant formula has attracted great interest. Very recently, de novo synthesis of 2′-FL in *P. pastoris* has been accomplished for the first time via homologous recombination-guided integration of the genes encoding the lactose transporter Lac12 and the relevant pathway enzymes into the genome of *P. pastoris* under the control of the constitutive P*_GAP_* promoter. By further optimizing the fermentation conditions, 2’-FL was produced at 0.276 g/L in a 5 L bioreactor [91].

#### 5.2.3. Polyketides

Polyketides are a class of widely distributed natural products produced during the secondary metabolism of plants or microorganisms, with rich chemical properties and unique physiological activities. Polyketide biosynthesis in *P. pastoris* was first reported in 2013 for the fungal polyketide compound 6-methylsalicylic acid (6-MSA) [92]. The genes encoding *Aspergillus nidulans* phosphopantetheinyl transferase (*npgA*) and *Aspergillus terrus* 6-MSA synthase (*atX*) were integrated into the genome of *P. pastoris* via homologous recombination, thus producing 2.2 g/L of 6-MSA in a 5 L bioreactor upon 20 h of methanol induction. On this basis, a more structurally complex polyketide citrinin was also synthesized in *P. pastoris* by assembling *npgA*, *Monascus purpureus* citrinin polyketide synthase gene *pksCT*, and several genes in the citrinin gene cluster [93]. In addition, the anti-hypercholesterolemia drug lovastatin and its precursor monacolin J were also efficiently synthesized from methanol in *P. pastoris* upon introduction and optimization of the pathway genes. By dividing the pathways into multiple modules and using a co-culture approach, monacolin J and lovastatin were produced at 593.9 mg/L and 250.8 mg/L, respectively [94].

#### 5.2.4. Other Compounds

The biosynthesis of other natural products in *P. pastoris* has also been attempted, such as flavonoids and stilbenes. Flavonoid synthesis relies on aromatic amino acids as precursors. A dominant *P. pastoris* chassis for the synthesis of L-tyrosine from glycerol was constructed by overexpressing key genes in the pentose phosphate pathway or shikimate pathway. This involved integration of an extra copy of endogenous *TKL1* (encoding transketolase), *SOL3* (encoding 6-gluconalactonase), *ZWF1* (encoding glucose-6-phosphate dehydrogenase) into the genome under the control of P*_AOX1_* via homologous recombination, and introduction and genomic integration of *ARO4^K229L^* (3-deoxy-D-arabino-heptulosonate-7-phosphate synthase) and *ARO7^G141S^* (chorismate mutase) from *S. cerevisiae* at the *AOX1* promoter locus driven by homologous recombination. This strain was further engineered to produce 1067 mg/L naringenin and 1825 mg/L eriodictyol via introduction of the relevant synthetic pathways, representing the most efficient production reported to date [96]. *P. pastoris* also supports the bio-production of alcohols and organic acids. Isopentanol biosynthesis requires 2-ketoisocaproate (2-KIC) as a key precursor. The accumulation of 2-KIC was promoted by overexpressing the endogenous valine and lysine biosynthesis pathways via an increase in the genomic copy numbers of the relative genes. Moreover, the keto-acid degradation pathway was expressed via genomic integration of *LlkivD* from *Lactococcus lactis* (encoding ketoisovalerate decarboxylase) and *ScADH7* from *S. cerevisiae* (encoding alcohol dehydrogenase 7) using homologous recombination, thus increasing the conversion of 2-KIC to isopentanol. Furthermore, *PDC* (encoding pyruvate decarboxylase) was deleted using CRISPR/Cas9 to reduce the formation of the byproduct ethanol. Finally, isopentanol was produced at 191 mg/L [98].

## 6. Conclusions and Prospects

*Pichia pastoris* has been developed to produce proteins for four decades and has emerged as a new chassis to produce diverse chemicals and natural products in recent years. With the unique property of utilizing methanol as a sole carbon source, metabolic engineering of *P. pastoris* is attracting increasing attention amid the great concern on global energy security, given the potential of methanol as a supply of energy and carbon source for biomanufacturing. *P. pastoris* has shown great success in the biosynthesis of some natural products with the titers reaching >1 g/L, indicating very promising applications for industrialization and commercialization. However, some issues are still present in *P. pastoris*, which hinder dedicated metabolic engineering of this microbe towards higher competence for on-demand production of chemicals and natural products. Firstly, extensive exploration of metabolic pathways is severely needed for their involvement in natural product biosynthesis, such as the mevalonate biosynthetic pathway, aromatic amino acid biosynthetic pathway, and so on. At present, the genome-scale model of *P. pastoris* is available but there is still very limited information on the details of metabolic flux distribution; in addition, more studies are required via the integration of multi-dimensional systems biology approaches with omics data and mathematical models to unveil potential gene targets responsible for the rearrangement of metabolic flux, especially for the accumulation of key precursors such as acetyl-CoA, malonyl-CoA, and dimethylallyl pyrophosphate. Secondly, enabling technologies and tools are not well developed. Although the CRISPR-based genome editing tool is available for *P. pastoris*, the current tools and genetic elements for metabolic engineering of *P. pastoris* are severely insufficient compared to its counterpart model strains including *S. cerevisiae*, *E. coli*, and other microorganisms. Future investigations are required to develop more tools such as promoter libraries, UTR sequences, terminator sequences, genomic sites for gene integration, dynamic regulation tools, and other genetic materials. Owing to the development of RNA-seq, single-cell analysis, and high-throughput analysis, it is convenient to develop tools for the use in genetic modification of *P. pastoris*. Moreover, artificial intelligence (AI) has been empowering metabolic engineering and synthetic biology, and construction of highly efficient strains can be accomplished precisely and efficiently compared to the routinely used time-consuming processes. All these technical achievements will definitely lead to an expanded spectrum of bio-products that can be synthesized at acceptable levels in engineered *P. pastoris* cell factories.

## Figures and Tables

**Figure 1 jof-09-01027-f001:**
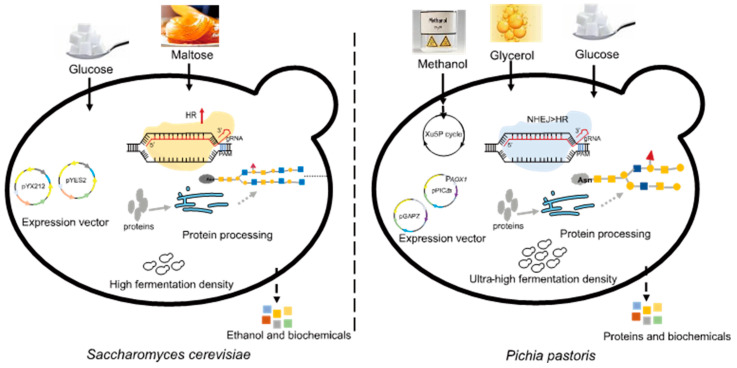
Differences between *Pichia pastoris* and *Saccharomyces cerevisiae*.

**Figure 2 jof-09-01027-f002:**
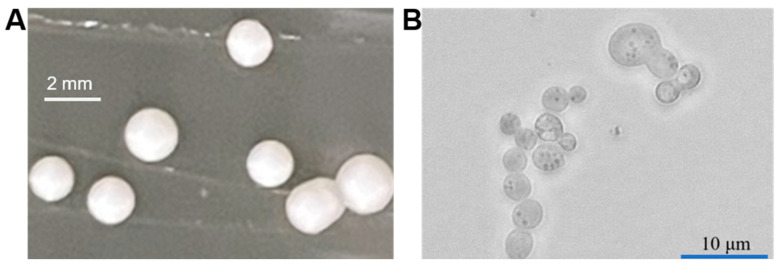
Colony morphology (**A**) and microscopic image (**B**) of *P. pastoris*.

**Figure 3 jof-09-01027-f003:**
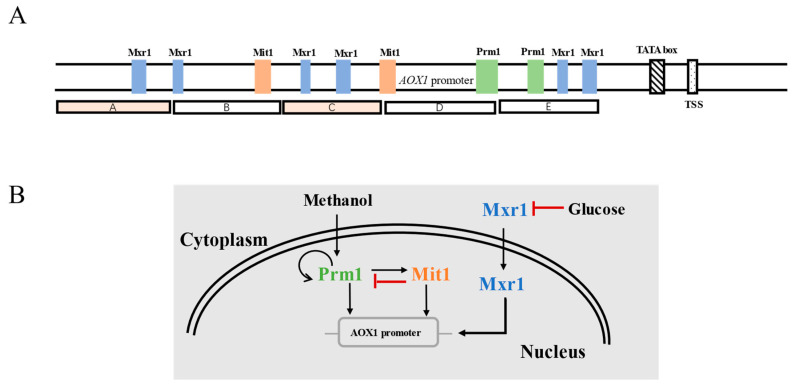
(**A**) The structure of P*_AOX1_*. (**B**) Transcriptional regulation of P*_AOX1_*.

**Figure 4 jof-09-01027-f004:**
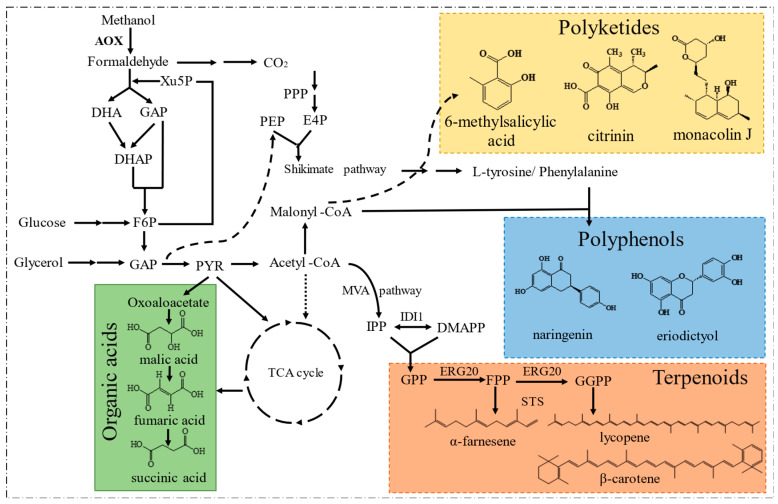
The portfolio of typical compounds produced by engineered *P. pastoris*.

**Table 1 jof-09-01027-t001:** Promoters used in *P. pastoris*.

Promoters	Regulation	Encoding Gene	Expression Level	Refs.
P_AOX1_	Induced by methanol	Alcohol oxidase	Strong	[36]
P_ADH2_	Induced by ethanol	Alcohol dehydrogenase	Stronger than P_GAP_	[51]
P_SNT5_	Induced by ethanol	synthetic promoter	Strong	[51]
P_CAT1_	Induced by methanol and oleic acid	Catalase	Similar to P_AOX1_	[52]
P_LAR3_	Induced by L-Rhamnose	L-rhamnose dehydratase	Similar to P_GAP_	[20]
P_LAR4_	Induced by L-Rhamnose	L-KDR aldolase	Weak	[20]
P_ADH3_	Induced by ethanol and methanol	Alcohol dehydrogenase	Strong	[53]
P_FLD1_	Induced by methanol and methylamine	Formaldehyde dehydrogenase	Similar to P_GAP_	[54]
P_PEX8_	Induced by methanol	Peroxisomal matrix protein	Weak	[55]
P_GTH1_	Induced by glucose	High-affinity glucose transporter	Similar to P_AOX1_	[56]
P_DAS1_	Induced by methanol	Dihydroxyacetone phosphate	Strong	[57]
P_FDH1_	Induced by methanol	Formate dehydrogenase 1	Strong	[49]
P_ICL1_	Induced by ethanol	Isocitrate lyase	~10% of P_AOX1_	[50]
P_CUPI_	Induced by glucose	Copper-binding metallothionein protein	~10% of P_AOX1_	[50]
P_GAP_	Constitutive	Glyceraldehyde-3-phosphate dehydrogenase	Strong	[38]
P_GCW14_	Constitutive	Glycosylphosphatidylinositol	Strong	[58]
P_TPI_	Constitutive	Triose phosphate isomerase	~4% of P_GAP_	[50]
P_PMA_	Constitutive	Plasma membrane (H^+^) ATPases	~81% of P_GAP_	[50]

**Table 2 jof-09-01027-t002:** Synthesis of various compounds in *P. pastoris*.

Classification	Products	Titer	Synthetic Strategies	Culture Conditions	Ref.
Terpenoids	*α*-Santalene	21.5 g/L	Overexpression of *tHMG1* and *IDI1*, increase in *SAS* copy number	Glucose/Fermenter	[85]
	Catharanthine	2.57 mg/L	Optimization of pathway genes and integration sites	Glucose/Fermenter	[86]
	Lycopene	6.146 mg/g	Promoter selection, increase in precursor supply and GGPP accumulation	Methanol/Shake flask	[87]
	α-Farnesene	3.09 g/L	Pathway reconstruction for the synthesis of NADPH and ATP	Glucose/Shake flask	[88]
Polysaccharides	Hyaluronic acid	0.8–1.7 g/L	Induction by P*_AOX2_*, low-temperature cultivation	Glucose/Fermenter	[89]
	Chondroitin sulfate	2.1 g/L	Codon optimization, endogenous promoter analysis, enhancement of the PAPS synthesis pathway	Methanol/Fermenter	[90]
	2′-Fucosyllactose	0.276 g/L	*GAP* promoter control	Glucose/Fermenter	[91]
Polyketides	6-Methylsalicylic acid	2.2 g/L	Heterologous expression of *atX* from *Aspergillus terreus* and *npgA* from *Aspergillus nidulans*	Methanol/Fermenter	[92]
	Citrinin	0.6 mg/L	Heterologous expression of *pksCT* from *Monascus purpureus* and *npgA* from *Aspergillus nidulans*, introduction of *mpl6* and *mpl7* from *Monascus purpureus*	Glucose/Shake flask	[93]
	Monacolin J	593.9 mg/L	Optimization of pathway gene expression, co-culture	Methanol/Fermenter	[94]
Flavonoids	Baicalein	401.9 mg/L	Ethanol induction, modularization of metabolic pathway	Ethanol/Shake flask	[95]
	Oroxylin A	339.5 mg/L	Ethanol induction, modularization of metabolic pathway	Ethanol/Shake flask	[95]
	Naringenin	1067 mg/L	Overexpression of *ARO4*^K229L^ and *ARO7*^G141S^ for efficient production of L-tyrosine	Glycerol/Fermenter	[96]
Alcohols	Isobutanol	48.2 mg/L	Overexpression of heterologous xylose isomerase and endogenous xylulokinase, introduction of the isobutanol pathway	Glucose/Microfermenter	[97]
	Isopentanol	191.0 mg/L	Overexpression of the endogenous valine/leucine biosynthetic pathways, and artificial keto-acid degradation pathway	Glucose/Microfermenter	[98]
Organic acids	Malic acid	2.79 g/L	Knockout of *gpi* optimized methanol assimilation, optimization of nitrogen sources	Methanol/Shake flask	[99]

## Data Availability

Not applicable.

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
