# Peer review of "Advances in Metabolic Engineering of Pichia pastoris Strains as Powerful Cell Factories"

_jof, 2023, doi:10.3390/jof9101027_

Round 1
Reviewer 1 Report
“Advances in Metabolic Engineering of Pichia pastoris as Powerful Cell Factories”
Reviewer comments:
This study grouped some relevant informations on the general physiological properties of P. pastoris and the readily available genetic tools and elements are described. Moreover, the recent achievements in P. pastoris-based biosynthesis of proteins, natural products and other compounds are highlighted. The existing issues and possible solutions are also discussed for the construction of efficient P. pastoris cell factories. The information provided is very updated with 50 % of bibliographic references from the last 6 years.
The structure of the review is divided in 3 main topics (Physiological characteristics of Pichia pastoris; The Pichia pastoris expression system; Practical applications of P. pastoris as a cell factory). Each topic is well developed and with enough examples well shown and discussed.
According to what it has been mentioned, the review could be interesting and usefull for the international scientific community. I recommend its publication but before some minor issues, like the one listed below, must be solved.
Specific issues:
Lines 24: The authors say “recently ” must be eliminated because when you check the reference (1) it is from 2009, and that`s not so recently.
Line 28: At the end of the paragraph references must be provided.
Line 30: The reference provided (2) does not correspond with the sentence.
Line 61: The reference provided (12) does not correspond with the sentence.
Line 91: The reference provided (24) does not correspond to Inan et al.
Author Response
Thank you for the critical comments. Please see the attachment for the revision.

Reviewer 2 Report
In this review, the authors give at first some general information about P. pastoris, describe different promoters used for expression in this yeast and describe the CRISPR/Cas9-mediated genome editing in P. pastoris. After that some examples on recombinant protein expression and production of value-added compounds were given. The manuscript is well written, but several aspects have to be described more in detail before publication.
Major comments:
Regarding AOX1 promoter: information on construction and utilization of mut+/mutS/mut- strains are missing and should be included.
Concerning signal peptides: foreign signal peptides are to some extent also accepted by P. pastoris. This statement has to be added and some examples given.
Concerning expression vectors: some characteristics (marker gene, integrative vs. episomal) of expression vectors need to be added and discussed. Additionally, a comparison of homologous gene integration vs. CRISPR/Cas9-mediated genome editing in P. pastoris is missing.
Furthermore, an overview on commonly used P. pastoris strains and their characteristics is missing.
Chapter 4.1.2 Human proteins: Some informatino on production of industrially produced insulin and interferon-α (as mentioned in the introduction) should be included.
According to the title “Advances in Metabolic Engineering of Pichia pastoris” chapter 4 should be divided in a section describing the achievements in heterologous (over-)production of proteins and a main section describing in detail the advances in metabolic engineering. The latter one should include information on how overexpression of certain genes involved in the metabolic pathway was achieved. Was it by manipulation of the intrinsic promoter or by integration of additional copies of the gene, or by overexpression of a foreign gene, etc.? For example, in line 398 it was only mentioned that “By overexpressing key genes in the pentose phosphate pathway or shikimate pathway...”, but no further details were given, which genes were involved and how exactly overexpression was achieved. However, this information would provide a deeper inside in how metabolic engineering in P. pastoris is conducted and would significantly improve the quality of the manuscript.
In general, additional information on how the respective genes have been integrated into the genome (via homologous recombination or CRISPR/Cas9) is missing and should be added.
Minor remarks
Fig. 1: strain names in italics
page 3, line 82: AOX1/2 stands for alcohol oxidase
Author Response

(The authors gave the same response as above.)

Reviewer 3 Report
The review attempts to cover the recent advances in metabolic engineering in Pichia pastoris. The authors tried to compile latest developments related to metabolic engineering strategy, regulatory pathways and recombinant protein production.
The review is written well but it does not add much information (or significance) to the information that already exist in the field covered by other reviews. The topics are not covered in detail or limited information is provided. For example, CRISPR/Cas9 technology, role of system biology, ALE have not been discussed in detail.
There are plethora of reviews available in Pichia that came within past year which discusses more or less similar topics (https://doi.org/10.1016/j.engmic.2023.100094, https://doi.org/10.1016/j.jobab.2023.01.007, https://doi.org/10.1007/s12033-023-00803-1, https://doi.org/10.3389/fmicb.2022.1059777).
Considering the similarity of this review with other reviews, I am not convinced if this review is adding much information that is not available in public domain to advance Pichia research.
Author Response

(The authors gave the same response as above.)

Round 2
Reviewer 3 Report
The authors have provided adequate responses to the questions/concerns raised by me in my previous review. I am satisfied with their responses. Therefore, I endorse the manuscript for publication in its current form.
Author Response
Thank you very much!